# Promotion of Mental Health Literacy in Adolescents: A Scoping Review

**DOI:** 10.3390/ijerph18189500

**Published:** 2021-09-09

**Authors:** Joana Nobre, Ana Paula Oliveira, Francisco Monteiro, Carlos Sequeira, Carme Ferré-Grau

**Affiliations:** 1Health School, Polytechnic Institute of Portalegre, 7300-555 Portalegre, Portugal; paulaoliveira@ipportalegre.pt (A.P.O.); franciscomonteiro@ipportalegre.pt (F.M.); 2Corporate Public Entity, Local Health Unit of North Alentejo, 7300-126 Portalegre, Portugal; 3Faculty of Nursing, University of Rovira i Virgili, 43003 Tarragona, Spain; carme.ferre@urv.cat; 4Group Inovation & Development in Nursing (NursID), Centro de Investigação em Tecnologias e Serviços de Saúde (CINTESIS), 4200-450 Porto, Portugal; carlossequeira@esenf.pt; 5Nursing School of Porto, 4200-072 Porto, Portugal

**Keywords:** adolescent, health literacy, health promotion, mental health, schools

## Abstract

In recent years, there has been an important commitment to the development of programs to promote mental health literacy (MHL) among adolescents, due to the prevalence of mental health problems and the low level of MHL that affects this group. The aim of this study was to map the structure and context of programmes/interventions for promoting MHL among adolescents in school settings. A scoping review was conducted following the guidelines of The Joanna Briggs Institute. We searched for studies on programmes/interventions promoting at least one of the components of MHL of adolescents, written in Portuguese, English or Spanish, published from 2013 to 2020, in MEDLINE, CINAHL Plus with Full Text, SciELO, SCOPUS, OpenGrey, RCAAP and in the article reference lists. This review included 29 articles. The majority of programmes/interventions addressed one or more of the four components of MHL, with the knowledge of mental disorders and stigma reduction components being the most covered; were taught by adolescent’s regular teachers; used face to face interventions; had a height variable duration; used non-validated instruments; were implemented in a classroom environment; and showed statistically significant improvements in adolescent’s MHL levels. More research is needed to implement/construct programmes/interventions promoting adolescents’ MHL concerning knowledge on how to obtain and maintain good mental health.

## 1. Introduction

The world is currently facing a very challenging public health problem: the significant prevalence of mental health problems in the general population and adolescents and young people [1,2], as well as their low/moderate levels of mental health literacy [3,4,5].

Mental health problems account for 12% of illnesses worldwide, and in developed countries, the figure rises to 23% [6]. As far as children and adolescents are concerned, around 10–20% are affected by these types of problems worldwide [7,8], with most of these problems onsetting during early adulthood and adolescence [9]. The first episode may occur before the age of 14 [8], with about half of the cases that appear throughout life appearing to settle at this age, as reported by Kessler’s study in 2005 [10].

The literature so far shows us that the levels of mental health literacy (MHL) of the general population and adolescents have been progressively increasing but are still at low/moderate levels [3,4,5]. This contributes to the absence of help seeking by adolescents, affects their development and increases the risk of psychiatric disorders recurring [11,12,13].

The concept of MHL is not recent. It emerged in the late 1990s through the investigations of Jorm and colleagues [14]. They defined it as the knowledge and beliefs about mental disorders that aid their recognition, management and prevention. Since then, researchers worldwide have shown a growing interest in this phenomenon (MHL), leading to the evolution of the definition of the concept. Currently, MHL refers to the knowledge and skills needed to foster mental health [15]. MHL has four components: understanding how to achieve and maintain good mental health, understanding mental disorders and their treatments, decreasing the stigma related to mental disorders and increasing the effectiveness of help seeking [16,17].

In this review, we adopted the WHO definition of mental health [18], which conceptualizes it as something more than the absence of disease; rather, it considers that it “is a state of well-being in which an individual realizes his or her own abilities, can cope with the normal stresses of life, can work productively and is able to make a contribution to his or her community” (p. 38). Another concept that is important to define is mental disorders, which encompasses several mental problems “generally characterized by some combination of abnormal thoughts, emotions, behaviour and relationships with others” [18] (p. 38). Regarding stigma, in this review, it is understood as “a mark of shame, disgrace or disapproval which results in an individual being rejected, discriminated against, and excluded from participating in a number of different areas of society” [19] (p. 18). We consider that the concept of help seeking “is an adaptive coping process that is the attempt to obtain external assistance to deal with a mental health concern” [20] (p. 180), including formal (health professionals, etc.) or informal sources (friends, family, etc.), encompassing not only the self-help strategies but also the first aid skills to support others [15]. It is also important to clarify that in this review, the authors consider that knowledge on achieving/maintaining good mental health comprises how to prevent mental disorders and promote mental health, such as having stable friendships and family support, to sleep enough, practice exercise, think in a positive way, avoid substance abuse, to have meaningful and enjoyable activities and relax [15]. According to the World Health Organization [8], adolescence encompasses all individuals aged between 10 and 19 years. It is the period in the life cycle between childhood and adulthood, characterised by profound physical and mental changes, during which attitudes develop and can still be changed [21,22]. Therefore, adolescence is considered a crucial period of opportunity to promote mental health [18]. Better literacy at a young age has a direct and positive impact on adult life. It enables adolescents to acquire the knowledge and define the attitudes and behaviours that will accompany them in their future lives [7,23]. Specifically, it gives adolescents the ability to positively manage their thoughts and emotions to build healthy social and family relationships, all based on a strong, positive sense of identity. Therefore, without a good level of MHL, adolescents will not develop healthily as they grow to adulthood [7,17], because without the knowledge and skills necessary to prevent the onset of mental disorders and to promote good mental health, these disorders are more likely to set in during adolescence and perpetuate themselves chronically. For this reason, adolescents are a primary target population for the promotion of MHL.

The World Health Organization [18] defined in its Mental Health Action Plan 2013–2020 that one of the objectives to attain at a global level is to implement strategies for the promotion and prevention of mental health problems, highlighting the importance of intervening not only on the needs of people with defined mental disorders but also on the protection and promotion of the mental health of all citizens. One such strategy is mental health literacy.

Given the significant prevalence of mental health problems in adolescence and low/modest levels of MHL, there is a need to explore the currently available evidence regarding programmes/interventions to promote MHL among adolescents. To this end, we chose to perform a scoping review which we considered to be the most appropriate methodology, given the objective of this type of review: to map the existing evidence in relation to a particular area or topic; to assess the reliability, relevance and potential costs of conducting a systematic literature review; to provide a synthesis of research findings and disseminate them; and to identify potential gaps in the existing literature [24,25,26,27].

After a preliminary survey was conducted in September 2019 in the JBI Database of Systematic Reviews and Implementation Reports, the Cochrane Database of Systematic Reviews, the CINAHL and in MEDLINE (via EBSCO); two systematic reviews of the literature were found in this area [23,28]. The systematic review by Wei et al. [28] included 27 articles published between 1988 and 2010. The authors concluded that there is little evidence of the effectiveness of programmes promoting MHL in schools. However, the interventions studied seemed promising as they showed positive results in the three outcomes studied (knowledge, attitudes/stigma and help-seeking behaviours). Concerning the systematic review by Morgado and Botelho [23], this included three studies, published between 2008 and 2012, with the authors concluding that cognitive–behavioural intervention, psycho–educational intervention and educational intervention are promoters of MHL and that school is the best means for promoting MHL, leaving as a future recommendation the importance of developing interventions in this area that are previously validated through pilot studies and then implemented more comprehensively.

Because in recent years, investment in developing programmes promoting MHL in adolescents has taken place, we felt the need to carry out this new scoping review to explore the existing evidence, from 2013, regarding adolescents’ MHL-promoting programmes/interventions, and to understand the characteristics of these programmes and the barriers/facilitators to their implementation, seeking to include published and unpublished studies.

This scoping review aims to map the structure and context of programmes/interventions for promoting MHL among adolescents in school settings, both at the level of published academic literature and grey literature.

The following primary research question was formulated to guide this study:What are the programmes/interventions for promoting MHL among adolescents in school settings?

In addition to this, the following secondary research questions were posed:What are the characteristics of the programmes/interventions for promoting MHL among adolescents highlighted in the literature?In what settings/contexts are these programmes/interventions carried out?What are the barriers and facilitators to the implementation of these programmes/interventions?

## 2. Materials and Methods

This scoping review follows the guidelines of The Joana Briggs Institute [26,27]. We used the checklist PRISMA-ScR (Preferred Reporting Items for Systematic reviews and Meta-Analyses extension for Scoping Reviews) for writing the review report [29].

The scoping review protocol was registered in the Open Science Framework on 10 November 2019 and is available for consultation [30].

### 2.1. Inclusion Criteria

Taking into account the questions formulated to guide this scoping review and using the PCC strategy—Participants, Concept, Context [26,27]—the following inclusion criteria were defined:Participants—articles targeting adolescents aged between 10 and 19 years, without diagnosed mental illness;Concept—studies on programmes/interventions for promoting MHL, covering at least one of the components of MHL;Context—we accepted studies that included adolescents in a school setting (2nd and 3rd cycles of basic education and secondary education, which corresponds to 5–12th grade), including online intervention and/or face to face intervention.

Concerning the types of studies, published and unpublished primary and secondary studies were included in this review to access a wider range of available information. We included studies written in Portuguese, Spanish or English, since these are languages in which the reviewers are proficient. We considered studies published from 2013 to 2020 to have only articles with the most recent evidence.

### 2.2. Search Strategy

As defined in the guidelines of The Joanna Briggs Institute [26,27], this scoping review was conducted in three stages.

In the first stage, an initial search limited to two electronic scientific databases was conducted (MEDLINE and CINAHL Plus with Full Text), using MESH (Medical Subject Headings) descriptors in the following Boolean phrase: (adolescent * AND ‘mental health’ AND literacy AND ‘health literacy’ AND program * AND nursing). This search was followed by an analysis of the terms used in the titles and abstracts of the articles found to identify all relevant terms associated, and to define the final Boolean phrase: (adolescent * AND ‘mental health’ AND (literacy OR ‘health literacy’ OR ‘mental health literacy’) AND (program * OR course * OR intervention *) AND promotion AND school *), where all terms are MESH terms except for ‘mental health literacy’, course*, intervention * and promotion, which are words from the general language.

It should be noted that, at this stage, we needed to introduce two small changes to what we had planned in the protocol of this scoping review. Specifically, we had to remove the term ‘nursing’ from the search strings since we found in the various search attempts that it could be reductive to the search, since we were exploring the existing programmes/interventions. The other change was to add four natural language terms suggested by the databases consulted (mental health literacy, course, intervention and promotion).

In the second stage, we searched the electronic scientific databases MEDLINE, CINAHL Plus with Full Text, SciELO, and SCOPUS, using the final Boolean phrase defined in the previous step: (adolescent * AND ‘mental health’ AND (literacy OR ‘health literacy’ OR ‘mental health literacy’) AND (program * OR course * OR intervention *) AND promotion AND school *), retrospectively from 1 January 2013 to 31 July 2020. In the electronic repositories OpenGrey (a European repository) and RCAAP (the Open Access Scientific Repository of Portugal), the search was carried out using a shorter Boolean phrase: adolescent * AND ‘mental health’ AND school *, using the same period, and MESH and DECS (Descriptors in Health Science) terms as descriptors. The search in both databases and repositories was conducted in December 2019 and updated in August 2020 (Table 1 and Table 2).

In the third stage, the reference lists of all articles included in the second stage were analysed, and additional relevant articles were identified and included in this scoping review.

### 2.3. Selection of the Studies

The studies obtained were imported and processed using the bibliographic reference management software Mendeley Desktop^®^ version 1.19.4. (Elsevier, Amsterdam, Netherlands) and Microsoft^®^ Excel 365 (Microsoft Corporation, Redmond, WA, USA).

The selection process consisted of two levels of screening of the articles obtained: (1) a review of the title and abstract and (2) a review of the full text.

The article selection process was carried out independently by two researchers, considering the previously defined eligibility criteria. In situations of disagreement between the researchers, the intervention of a third researcher was requested to reach a consensus. The full text was reviewed in cases in which the title and abstract did not contain sufficient information for an adequate decision.

### 2.4. Data Extraction

Data were extracted from the articles with a full-text format that met the inclusion criteria, using an instrument created by the reviewers (Appendix B, Table A1), according to the model proposed by The Joanna Briggs Institute [26,27] and aligned with the objectives and questions of the review. Data extracted from the articles were as follows: author(s), year of publication, country, objective(s) of the study, study design, participants, characteristics of the programmes/interventions implemented, data collection instruments used, main outcomes and barriers/facilitators. Any disagreements between the reviewers were resolved through discussion or with the use of a third-party investigator.

## 3. Results

In the beginning, 104 articles were found in the search in the four databases and the two repositories consulted. After removing the duplicates and applying all the procedures, 29 articles were obtained. The results of the article selection process are summarised in Figure 1 in a PRISMA diagram [31].

The list of included studies and the description of their characteristics are shown in Appendix A. The studies included according to the components of MHL are in Table 3.

The articles included in the review were published from 2013 to 2020. Seven articles were published in 2016, five articles in 2014, five articles in 2018, four articles in 2015, three articles in 2013, two articles in 2019, two articles in 2020 and one article in 2017.

Of the 29 articles included, twelve were experimental studies (of which two were study protocols and two were pilot studies), nine were quasi-experimental studies (two of which were pilot studies), three were descriptive articles, two were secondary analyses, two were systematic reviews of the literature and one was a mixed study (pilot study).

### 3.1. Component—Knowledge on Achieving/Maintaining Good Mental Health

Of the eight articles addressing knowledge on how to obtain/maintain good mental health, two were experimental studies [32,33], two were quasi-experimental [34,35], two were descriptive articles [36,37] and two were secondary analyses [38,39].

The participants in the programmes/interventions were adolescents aged 10 to 18 years. In five of those programmes, the adolescents were aged ≤14 years.

The duration of the programmes/interventions in these eight studies ranged from a single 3 h session to multiple sessions that could run up to a total of approximately 24 h.

The assessment instruments used were mostly developed by the authors of the programmes/interventions (*n* = 4), followed by the combined use of validated instruments with instruments developed by the authors (*n* = 2) and the use of validated instruments (*n* = 1). One of the studies did not mention the instruments used.

After examining the assessment moments, we found that all the studies assessed the programmes/interventions at baseline (*n* = 8), five performed the assessment immediately after, one (*n* = 1) performed the assessment after 2 weeks and one (*n* = 1) performed the assessment after 3 months. A follow-up stage was mentioned in five studies. Two studies implemented a follow-up at 2 months, two studies at 6 months and one study at 6 and 12 months.

Regarding the results, all studies referred to increased knowledge, but upon close examination, they only assessed knowledge about mental disorders.

It is worth noting that four of these articles referred to the same programme (“The Guide”) implemented in the same country (Canada). Still, the samples were different in terms of the ages of the participants or the country’s regions.

### 3.2. Component—Knowledge about Mental Disorders and Their Treatments

Twenty-four articles addressed programmes/interventions that aim to promote knowledge about mental disorders and their treatments, of which ten were experimental studies [32,33,40,41,42,43,44,45,46,47], seven were quasi-experimental studies [34,35,48,49,50,51,52], three were descriptive articles [36,37,53], two were secondary analyses [38,39], one was a mixed study [22] and one was a systematic literature review [54].

The participants in these programmes/interventions were adolescents whose ages ranged from 10 to 18 years, with most studies targeting adolescents aged 14 years or younger.

Regarding the programme/intervention duration, there was wide variability, from a single 45 min session to multiple sessions. Only two studies did not mention the duration of their programmes/interventions.

The programmes/interventions in this component of MHL used mostly assessment tools developed by the authors (*n* = 10). Other programmes/interventions used validated instruments (*n* = 7), or combined validated and own instruments (*n* = 4). Only three studies did not mention the instruments used.

Three studies did not mention any information concerning the assessment moments. Of the remaining twenty-one, all were assessed at baseline, thirteen were assessed immediately after the intervention, three were assessed after 3 months, two studies were assessed after 2 weeks, two studies were assessed after 1 week and one study was assessed after 6 weeks. The follow-up period was included in thirteen studies, of which four studies at 6 months, three studies at 3 months, two studies at 2 months, two studies at 6 and 12 months, one study at 4 months and one study at 12 and 24 months. It should also be noted that most studies present 3 moments of assessment (*n* = 10), followed by those with 2 moments (*n* = 8) and with 4 moments (*n* = 3).

In terms of the results, most programmes/interventions report increased knowledge about mental disorders and their treatments (*n* = 18), the results of which are statistically significant, and only one study reports that the increase in knowledge was slight, in which the results are not statistically significant [47]. Some studies that contemplate this category did not refer to the related results (*n* = 5).

### 3.3. Component—Reducing Stigma Associated with Mental Disorders

Of the 24 articles that address programs/interventions whose objective is to reduce the stigma, ten were experimental studies [32,33,42,43,44,45,46,47,55,56], six were quasi-experimental [35,48,49,50,57,58], three were descriptive articles [36,37,53], two were secondary analyses [38,39], two were systematic reviews of the literature [54,59] and one is a mixed study [22].

The participants of the studies encompassed in this MHL category were aged 10 to 18 years, with most of the studies targeting adolescents aged ≤14 years.

The duration of these programmes/interventions ranged from a single 10 min session to multiple sessions, up to 4 months. Only two studies did not mention the duration of their programmes/interventions.

Most studies used validated assessment instruments (*n* = 10) to assess the programmes/interventions. Other studies used their own instruments (*n* = 8) or a combination of validated and their own instruments (*n* = 3). Only three studies did not mention the instruments used.

In four studies, the time points of intervention assessment were not mentioned. All the remaining twenty studies assessed the programmes/interventions at baseline. Twelve studies assessed the programmes/interventions immediately after the intervention; three assessed after 3 months, two studies after 2 weeks, two studies after 1 week and one study after 6 weeks. A follow-up period was contemplated in eleven studies. Of these, three studies implemented a follow-up at 6 months, two studies at 3 months, two studies at 2 months, two studies at 1 month and one study at 6 and 12 months.

Most of the programmes/interventions (*n* = 19) achieved a reduction in the stigma associated with mental disorders, and two studies did not register any change after implementing the programme/intervention. Those two studies were also the ones whose results were not statistically significant [45,57]. Three studies did not refer to the results of this component.

### 3.4. Component—Help-Seeking

Fifteen articles addressed programmes/interventions that aimed to promote help-seeking, of which seven were experimental studies [32,40,41,43,44,46,47], five were quasi-experimental studies [35,48,50,51,52], one was a descriptive article [53], one was a mixed study [22] and one was a systematic literature review [54].

The participants in the programmes/interventions were adolescents aged 10 to 18, thirteen of which were aged ≤14 years.

The duration of the programmes/interventions promoting help-seeking varied from a single session to multiple sessions. Two studies did not mention the duration of their programmes/interventions.

The programmes/interventions in this component of the MHL used mostly assessment tools developed by the authors (*n* = 7). Others used validated instruments (*n* = 4) or the combination of validated and own instruments (*n* = 2). Two studies did not mention the instruments used to assess this component.

Two studies did not mention any information concerning the assessment moments of the programmes/interventions. All the remaining thirteen studies assessed the programmes/interventions at baseline. Six assessed immediately after the intervention, three assessed after 3 months, two studies after 2 weeks and two studies after 1 week. A follow-up period was included in eight studies. Four studies implemented a follow-up at 3 months, two studies at 6 and 12 months, one study at 12 and 24 months and one study at 6 months. It should also be noted that most studies presented two moments of assessment (*n* = 6), followed by those with three moments (*n* = 4) and with four moments (*n* = 3).

Most programmes/interventions reported increased help seeking (*n* = 11), the results of which were statistically significant, and only one study showed results that were not statistically significant [47]. Some studies that contemplated this component did not have results available (*n* = 4).

In general, the following aspects were indicated as barriers to the implementation of the programmes/interventions common to all the MHL components in the articles included: the short duration of the intervention (*n* = 1), the use of English instead of the native language (*n* = 1), the difficulty in coordinating the implementation of the programme/intervention with the various stakeholders in the school (*n* = 1), the programme/intervention interrupting the school curricula (*n* = 1) and the lack of incentives for the participants (*n* = 1). On the other hand, the following aspects were mentioned as facilitators: not having to resort to staff from outside the school (*n* = 6), requiring only existing school resources (*n* = 4), the programme/intervention being administered as part of the school curriculum (*n* = 3), students being active agents of the intervention (*n* = 1), the use of staff from outside the school (*n* = 1), the use of role-playing rather than direct contact with people with mental illness (*n* = 1), the inclusion of a quiz at the end of the programme/intervention (*n* = 1), the incorporation of yoga exercises and postures (*n* = 1), being a concise programme/intervention (*n* = 1), being a short programme/intervention (*n* = 1) and being a quick programme/intervention without any associated expenses (*n* = 1).

## 4. Discussion

This review provides a comprehensive synthesis of the available evidence on the programmes/interventions promoting MHL in adolescents in school settings.

The first research question of this review intended to know what the programmes/interventions are for promoting MHL among adolescents in school settings. The results of this review show that most programmes/interventions address one or more of the four components of MHL defined by Kutcher, Wei and Coniglio [16]; that is, mental health disorders and problems, signs/symptoms and treatments, myths related to mental illness, non-stigmatising attitudes/behaviours and options/sources for help seeking. However, the programmes/interventions that seek to intervene in the component related to knowledge on how to obtain and maintain good mental health fall short of what is required. Therefore, future research should develop programmes/interventions with a more salutogenic and positive perspective regarding the MHL of adolescents. This scoping review highlights this gap, aligning with what is known from previous research [17].

Regarding the second research question, the objective was to discriminate the characteristics of programmes/interventions that promote MHL among adolescents. Most programmes/interventions targeted adolescents aged ≤14 years, thus making an important contribution to preventing the onset of mental health problems at an early age [8].

About half of the programmes/interventions were taught by the adolescents’ regular teachers. The rest used staff from outside the school, with only a few being taught by health professionals. These results highlight the need for greater intervention from health professionals, particularly those in primary health care and specifically nurses, who play a decisive role in the community’s health [5]. Nurses know the needs and specificities of their community like no one else, and this knowledge enables them to intervene holistically. Considering health professionals’ competences and level of expertise, we believe that one of the future options in this field may be a more active intervention by nurses and other health professionals, both in the implementation and administration teams of the programmes that promotes adolescents’ MHL, as well as in the teachers’ education/training on these programmes/interventions.

In terms of the strategies used, the results show the use of expositive, demonstrative, participative methodologies based on contact (direct or indirect) and/or the supply of information material. These strategies were used in isolation or as complements. In most studies, complementarity proved to be an added value in achieving an increase in the MHL of adolescents. However, one study showed that adding contact with patients with mental disorders did not add value to the educational intervention [47].

The variability of the duration of the programmes/interventions analysed indicates that they may be flexible in terms of time, even though a significant proportion of the analysed programmes/interventions state that the fact that they are of short duration is an advantage because they save resources. However, while it is true that when the aim is to intervene at the level of knowledge and help-seeking behaviour, a short-term intervention is effective, it is also true that when the objective is to act on attitudes, it is probably better to opt for a longer intervention, since attitudes cannot be changed easily, and they need time to be internalised and sedimented at a cognitive, emotional and behavioural level [34]. It is also suggested that in the future, programmes/interventions should have follow-up periods not only in terms of assessments of their short- and long-term effects, as occurred in a significant part of the studies included in this review, but also in terms of booster sessions, as in the study carried out by Lubman et al. [40], as the literature indicates their importance in increasing and maintaining the effects of interventions [60].

None of the reviewed studies used instruments to assess outcomes concerning knowledge about achieving and maintaining good mental health, which is in line with the findings of Wei et al. [9]. Future research should use instruments that assess this component of the MHL or, in its absence, should construct a new one. Furthermore, no study used an instrument that assessed the four components of MHL, probably because no instrument is considered a gold standard for assessing these components together, a situation already detected by Wei et al. [9]. The filling of this gap represents a future research area. Although about half of the programmes/interventions used validated instruments, a significant proportion used non-validated instruments, which compromises the appropriate assessment of results and the possibility of comparing them, a situation also mentioned by Wei et al. [28].

The third research question intended to know in which settings/contexts these programmes/interventions were carried out. Most of the programmes/interventions were implemented in a classroom environment. This fact demonstrates the importance of the school setting in promoting the MHL of adolescents and is in line with the research reported in this area [17,23]. It is also important to mention that the most programmes included in this scoping review consist of face to face interventions, only two programmes encompass online interventions (“EspaiJove.net” e “The Guide and MyHealth Magazine”) [32,36] and only one compares the same programme in its face to face version with the online version (“StresSOS”) [41]. Both “EspaiJove.net” [32] and the “StresSOS” [41] programmes do not have results yet because they are study protocols, but “The Guide and MyHealth Magazine” [36] already has results and they indicate improvements in the adolescents’ MHL when combining face to face and online interventions. However, we believe that in the future, more studies will be needed to compare both interventions and gather more evidence.

Finally, regarding the fourth research question, the objective was to know the barriers and facilitators to the implementation of these programmes/interventions. The results obtained indicate that the main barriers to implementing the programmes/interventions are the difficulty of coordination with the various school stakeholders, the interruption of school curricula and the lack of incentives for participants. The main facilitators were the programmes/interventions being part of the school curriculum, not depending on resources outside the school and using interactive methodologies. These aspects should be considered when implementing future interventions so as not to compromise their effectiveness.

Although this scoping review followed The Joanna Briggs Institute guidelines to maintain methodological and scientific rigour and was conducted by two independent researchers, it is possible to identify some limitations. First, the search was limited to articles published in Portuguese, English, or Spanish, which may have meant that important articles written in other languages were not included. Second, the quality of the included articles was not assessed, a situation inherent to the methodology of a scoping review, which prevents the presentation of recommendations for clinical practice. Thirdly, the fact that the original authors were not contacted to obtain information missing from the articles may have led to an inaccurate interpretation of the studies. Fourth, the fact that no studies were included in the scope of other areas (e.g., social sciences, etc.), nor articles with programmes implemented in contexts other than schools, is also a limitation.

## 5. Conclusions

The results of this review allow us to identify programmes/interventions that promote the MHL of adolescents, as well as to provide clues about some of the characteristics that such programmes/interventions should have, about some of the barriers and facilitators to their implementation and, finally, about the gaps found in this research area.

Although most of the analysed studies have apparently shown positive results in promoting the MHL of adolescents in school settings, these results are difficult to interpret and compare due to the lack of use of validated instruments and the great variability of the assessment instruments used.

Future research should be conducted to harmonise programmes/interventions that aim to promote each of the components of MHL, and MHL holistically in the adolescent population. To this end, further experimental or quasi-experimental studies should be carried out to obtain the best possible evidence, using validated assessment tools and including follow-up periods. Interventions should focus on adolescents aged ≤14 years; could be of short duration if the aim is to increase knowledge or help seeking, or of longer duration if the objective is to intervene at the level of adolescents’ attitudes/stigma; may include ‘booster’ sessions to reinforce and maintain the levels of MHL; should take place in the classroom; use complementary expository and interactive strategies; and have a more active intervention from health professionals.

We should focus on the implementation or construction of programmes/interventions that promote knowledge on how to obtain/maintain good mental health and the use or construction of instruments that assess this component of MHL, whose importance is currently being increasingly recognised by research.

## Figures and Tables

**Figure 1 ijerph-18-09500-f001:**
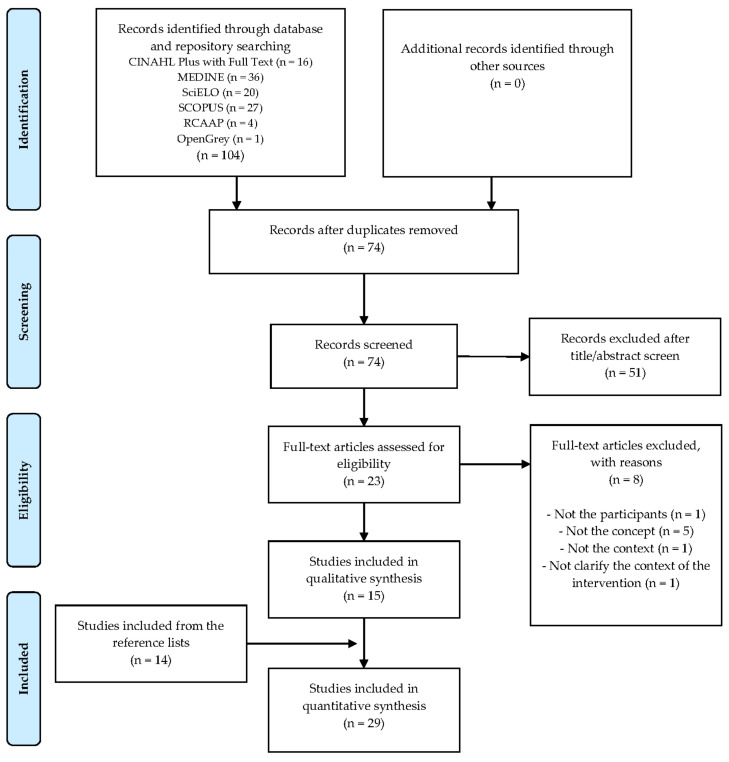
Article search and selection process—PRISMA diagram.

**Table 1 ijerph-18-09500-t001:** Studies obtained by search term and electronic database.

Search	CINAHL Plus with Full Text	Medline	SciELO	Scopus
S1: adolescent *	139,077	2,112,677	18,767	2,304,095
S2: “mental health”	145,021	279,261	10,078	331,070
S3: literacy	21,227	23,359	2542	88,214
S4: “health literacy”	7534	10,829	478	15,027
S5: “mental health literacy”	511	776	48	1146
S6: program *	511,350	1,398,162	65,511	3,505,488
S7: course *	122,063	616,864	19,770	1,511,044
S8: intervention *	458,783	1,024,234	32,994	1,467,357
S9: promotion	102,071	184,606	10,507	296,991
S10: school	173,702	4,315,891	32,732	1,182,107
S11: (S3 OR S4 OR S5)	21,227	23,359	2542	88,214
S12: (S6 OR S7 OR S8)	971,745	2,791,998	106,869	6,065,802
S13: (S1 AND S2 AND S11 AND S12 AND S9 AND S10)	18	42	34	34
With time limiter 2013–2020	16	36	20	27

* Search term with truncation.

**Table 2 ijerph-18-09500-t002:** Studies obtained by search term and repository.

Search	RCAAP	OpenGrey
S1: adolescent *	13,879	4001
S2: “mental health”	3081	1948
S3: school	13,587	23,518
S4: (S1 AND S2 AND S3)	4	17
With time limiter 2013–2020	4	1

Abbreviations: RCAAP, the Open Access Scientific Repository of Portugal. * Search term with truncation.

**Table 3 ijerph-18-09500-t003:** List of studies included in the review according to the components of MHL.

Author (s), Year	Country	Component: Knowledge Good MH	Component: Knowledge MH Disorders	Component: Stigma	Component: Help Seeking
Lubman et al. (2016) [40]	Australia		✓		✓
Yang et al. (2018) [57]	USA			✓	
Campos et al. (2014) [22]	Portugal		✓	✓	✓
Eschenbeck et al. (2019) [41]	Germany		✓		✓
Bella-Awusah et al. (2014) [34]	Nigeria	✓	✓		
Lindow et al. (2020) [48]	USA		✓	✓	✓
Hui et al. (2019) [49]	China		✓	✓	
Perry et al. (2014) [42]	Australia		✓	✓	
Casañas et al. (2018) [32]	Spain	✓	✓	✓	✓
Ojio et al. (2020) [53]	Japan		✓	✓	✓
Kutcher, Bagnell & Wei (2015) [36]	Canada	✓	✓	✓	
Mcluckie et al. (2014) [38]	Canada	✓	✓	✓	
Gonçalves et al. (2016) [55]	Portugal			✓	
Santos et al. (2013) [37]	Portugal	✓	✓	✓	
Campos et al. (2018) [43]	Portugal		✓	✓	✓
Mellor (2014) [59]	UK			✓	
Hart et al. (2016) [50]	Australia		✓	✓	✓
Hart et al. (2018) [44]	Australia		✓	✓	✓
Swartz et al. (2017) [45]	USA		✓	✓	
Schilling et al. (2016) [46]	USA		✓	✓	✓
Kutcher, Wei & Morgan (2015) [39]	Canada	✓	✓	✓	
Milin et al. (2016) [33]	Canada	✓	✓	✓	
Chisholm et al. (2016) [47]	UK		✓	✓	✓
Ojio et al. (2018) [51]	Japan		✓		✓
Ojio et al. (2015) [52]	Japan		✓		✓
Skre et al. (2013) [35]	Norway	✓	✓	✓	✓
Gonçalves et al. (2015) [56]	Portugal			✓	
Salerno (2016) [54]	USA		✓	✓	✓
Martínez-Zambrano et al. (2013) [58]	Spain			✓	

Abbreviations: MH, mental health; MHL, mental health literacy; UK, United Kingdom; USA, United States of America.

## Data Availability

Data sharing does not apply to this article as no new data were created or analysed in this study.

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
