# Peer review of "Promotion of Mental Health Literacy in Adolescents: A Scoping Review"

_ijerph, 2021, doi:10.3390/ijerph18189500_

Round 1
Reviewer 1 Report
Dear authors,
I attach the document with my reviewer's report.

Reviewer 2 Report
This is an interesting and well composed scoping review focusing on the promotion of mental health literacy in schools, which I enjoyed reading. It was welcome to see this addition to the adolescent mental health literacy literature, which builds on existing systematic reviews of MHL interventions delivered in school settings (e.g. Wei et al., 2013). The authors contribute to the literature with this scoping review by summarising key facilitators and barriers to effective delivery of MHL programmes in schools, and that MHL interventions should be developed which focus on all core component of MHL
Broad Comments:
The authors present an interesting and relevant review, which highlights the current state of programmes / interventions aiming to improve adolescent MHL, and thus increase effective help-seeking. The study identifies clear gaps in what is known in relation to delivery of MHL interventions across all components of MHL. It is particularly useful to the literature that the review was structured as such, as MHL lacks clarity in its conceptualisation and is often over dependent on improving recognition of psychiatric conditions, with other components of MHL overlooked. The authors, by choosing to frame their review and result by how well current programmes cover all components of MHL is particularly valuable in providing some needed insight into the landscape of MHL provision for adolescents, and mapping the delivery of interventions and highlighting in which components this needs improve. This was largely a well presented and robust review. I find this to be a well-considered and communicated study, which needs only some minor changes (discussed in greater detail below).
Specific comments:
- Introduction: Overall, I thought the introduction was well-written and informative. Rationale for the research, and key aims were clear, and good summaries of key terms and arguments were provided.
There are only minor changes I think should be made in this section:
Line 37: The language in the sentence “The first episode may occur before the age of 14”, felt a little tentative or vague, could the authors add a small note on the prevalence of those experiencing a first episode of poor mental health.
Lines 61 – 62: The authors state, “Therefore, without a good level of MHL, adolescents will not develop healthy as they grow to adulthood [7,16].” Can the authors expand on this in text? In what ways will adolescents not develop healthily into adulthood?
Lines 65-66: The authors state, "...one of the objectives to attain at a global level is to implement strategies for the 65 promotion and prevention of mental health, highlighting...". I believe the authors meat the 'prevention of mental health problems.
Lines 98-99: Can you comment here whether your aim specifically aimed to include interventions globally? I realise that countries are listed in Table 3, but if the aim was to capture global / 'western' countries or regions or otherwise that this should be clear in the aims.
2. Materials and methods
This section was very clear and informative in terms of reproducibility, changes in this section are very minor in nature.
Lines 125-126: Can the authors explain what is meant by "2nd and 3rd cycles of basic education'?
Lines 145-147: No change needed here, but wanted to flag this as a good example of transparency and justification around changes in protocol.
3. Results
Lines 200-201: I realise the authors here are aiming to start with the year with the most articles published, and progress to the year with the fewest articles published, but this may be more readable if done in chronological order (e.g. in 2013, three articles were published, in 2014...2015... etc). However this is very minor in nature and I leave this to the discretion of the authors.
Line 286: The phrasing of "The moments of assessment of the programmes/interventions were not mentioned in four studies", seems a little grammatical off? I understand that the authors are referring to time points at which the interventions were assessed (e.g. baseline, 12 weeks post intervention etc), but wonder if this could be expressed more clearly?
4. Discussion
I enjoyed reading the discussion section of this study, and believe that the finding that " future research should develop programmes/interventions with a more salutogenic and positive perspective regarding the MHL of adolescents" is an important finding which progresses the narrative on delivery of MHL in schools.
Again, suggested changes to this section are minor in nature:
Lines 359-360: Here the authors state that nurses would be ideal for delivering MHL in schools - although later in the discussion it is stated that a key facilitator of MHL delivery in schools is the ability to use existing resources. While a good case is made for why nurses / health professionals may be well placed to deliver MHL, I wonder if more consideration could be given to other alternatives (e.g. it is mentioned that about half of interventions were taught by teachers - were these successful or to be avoided? It may also be useful to consider adolescent peer delivery etc as an option. This is not an exhaustive list, but I feel that more evidence from the literature should be offered in balance.
5. Conclusions
I can offer no suggestions for improvements to this section - it summarises key findings and suggestions for future research comprehensively.
Reviewer 3 Report
Please see attached file. It may be useful to include other health care providers.
